# Challenge Tooth Regeneration in Adult Dogs with Dental Pulp Stem Cells on 3D-Printed Hydroxyapatite/Polylactic Acid Scaffolds

**DOI:** 10.3390/cells10123277

**Published:** 2021-11-23

**Authors:** Rung-Shu Chen, Sheng-Hao Hsu, Hao-Hueng Chang, Min-Huey Chen

**Affiliations:** 1Graduate Institute of Clinical Dentistry, School of Dentistry, College of Medicine, National Taiwan University, Taipei City 100229, Taiwan; d92548016@ntu.edu.tw (R.-S.C.); d95549005@ntu.edu.tw (S.-H.H.); 2Department of Dentistry, National Taiwan University Hospital, Taipei City 100229, Taiwan

**Keywords:** dental pulp stem cells, 3D printing, polylactic acid scaffolds, tooth regeneration

## Abstract

Tooth regeneration is an important issue. The purpose of this study was to explore the feasibility of using adult dental pulp stem cells on polylactic acid scaffolds for tooth regeneration. Three teeth were extracted from each side of the lower jaws of two adult dogs. In the experimental group, dental pulp stem cells were isolated and seeded in the 3D-printed hydroxyapatite/polylactic acid (HA/PLA) scaffolds for transplantation into left lower jaw of each dog. The right-side jaw of each dog was transplanted with cell-free scaffolds as the control group. Polychrome sequentially labeling was performed for observation of mineralization. Dental cone beam computed tomography (CBCT) irradiation was used for assessment. Nine months after surgery, dogs were euthanized, and the lower jaws of dogs were sectioned and fixed for histological observation with hematoxylin and eosin staining. The results showed that the degree of mineralization in the experimental group with cells seeded in the scaffolds was significantly higher than that of the control group transplanted with cell-free scaffolds. However, the HA/PLA scaffolds were not completely absorbed in both groups. It is concluded that dental pulp stem cells are important for the mineralization of tooth regeneration. A more rapid absorbable material was required for scaffold design for tooth regeneration.

## 1. Introduction

Tooth plays an important role in overall health and activity. Tooth loss or damage is quite frequent. Regenerative medicine has emerged as a novel therapeutic approach to promote regeneration in a more predictable manner. It is an attractive medical alternative to conventional treatment. With the increasing knowledge of stem cell biology, tooth regeneration is one of the ultimate goals of restoring the loss of natural teeth. Compared with the traditional oral restoration treatment, tooth regeneration has unique advantages and is currently the focus of oral biomedical research. Studies have indicated that cell-based strategies show promising potential for regenerating the whole tooth structure in rodents [1]. In our previous study, we had obtained tooth germ cells from mini pigs and implanted them in the alveolar bone of the original mini pigs after culture and have successfully grown teeth the same size as those of the mini pigs. The use of scaffolds can also regenerate structures that have dentin, cementum, and periodontal ligament as tooth roots [2]. Our studies have also shown that the characteristics of materials can regulate cell growth and differentiation [3], which has an important impact on the application for tissue regeneration. The biggest challenge was to find a simple and rapid method to control tooth development.

The study of dental pulp stem cells (DPSCs) by Gronthos’s group [4] reported an immunophenotype similar to that of bone marrow stem cells (BMSCs), along with the formation of a calcified nodule upon treatment with differentiation medium in vitro. This group transplanted DPSCs into the dorsal surface of immunocompromised mice with DPSCs have the capacity to differentiate into odontoblast-like cells. This differentiation capacity was revealed by the finding that DPSCs mixed with hydroxyapatite/tricalcium phosphate (HA/TCP) were able to regenerate a dentin/pulp-like structure by transplantation into immunocompromised mice, with the results showing that DPSCs were able to regenerate a dentin/pulp-like complex. The research about dentin-pulp regeneration had also been widely investigated [5,6,7,8]. The application of dental pulp stem cells for regeneration is promising [9].

Fused filament fabrication (FFF), also called fused deposition modeling (FDM) is one of the widely used 3D-printing techniques to fabricate the objectives composed of thermoplastic polymers and composites. Polylactic acid (PLA) is a US-FDA approved thermoplastic biodegradable polymer, but the acidic degradation product of PLA implants may cause non-bacterial inflammation of the surrounding tissue [10]. Hydroxyapatite (HA) has been intensively investigated and widely used in biomedical applications due to its bioactivity and biodegradability. HA, like other bioceramics, performed the neutralizing capacity to the acidic products from in vivo degraded biomedical polymers [11,12,13]. However, 3D printing of biomedical composite is always constrained by the capability of 3D printers and material processing equipment. Generally, a lab-scale extruder to prepare biomedical composite filaments of specific diameters was necessary for most FDM 3D printers to fabricate scaffolds [14]. There were several bioprinters capable of fabricating the scaffolds of various customized materials due to the variety of accessories [15]. However, both the bioprinters and the lab-scale extruders were expensive. Therefore, we printed the PLA scaffolds using a desktop FDM 3D printer, and then we coated HA on the scaffolds via the dip-coating process to obtain HA/PLA scaffolds.

We also found that by isolating DPSCs with a fluorescence-activated cell sorting system, about 1.22% STRO-1^+^ CD146^+^ cells from the mixture of isolated cells from eight teeth were sorted as DPSCs [16]. In this study, we tried to challenge the tooth regeneration in adult dogs with transplantation of dental pulp stem cells isolated from adult dogs and inserted in the 3D-printed PLA scaffolds.

## 2. Materials and Methods

### 2.1. 3D-Printed PLA Scaffolds

Architectural Design of 3D-printed scaffolds was conducted using the software Schetchup Make 2016 (Trimble Inc., Sunnyvale, CA, USA). Figure 1a–c shows the side view, top view, and the dimension of scaffolds. The 3D structures exported to STL files were sliced by Ultimaker Cura 2.7 (Ultimaker B.V., Utrecht, The Netherland) to create a g-code for 3D printing. The print speed was 30 mm/min; the layer height and line widths were 100 and 250 μm, respectively. The scaffolds designed for incisor and premolar regeneration were named the I-scaffold and P-scaffold, respectively.

An Ultimaker 2.0 Plus 3D printer equipped with a 250 μm copper nozzle and PLA Transparent filament (Ultimaker) of 2.85 mm diameter were utilized to conduct 3D printing. While printing, the build plate temperature and the nozzle temperature was set at 60 and 200 °C, respectively.

### 2.2. Dip-Coating HA on PLA Scaffolds

The dip-coating solution is composed of 70.0 wt.% tetrahydrofuran (THF, HPLC grade, Thermal Fisher Scientific Inc., Waltham, MA, USA), 25.0 wt.% ethyl alcohol (EtOH, 99.5%, Shimakyu Co. Ltd., Osaka, Japan), 4.5 wt.% hydroxyapatite nanopowder (HA, <200 nm, Sigma-Aldrich, St. Louis, MO, USA), and 0.5 wt.% polylactic acid cut from the PLA Transparent filament. To prepare 100 g dip-coating solution, 0.5 g PLA was completely dissolved in 70 g THF. Then, 25 g EtOH and 4.5 g HA powder were poured into PLA/THF solution and a HA dip-coating solution was obtained.

The HA dip-coating process was conducted by a KSV Dip-Coater. The 3D-printed sample was dipped into the dip-coating solution at a rate of 200 mm/min. After 10 s, the sample was withdrawn at a rate of 200 mm/min, and then the dip-coated scaffolds were placed on Kimwipes (Kimtech Science, Kimberly-Clark, Irving, TX, USA) for 10 min to remove the excess dip-coating solution. Finally, the HA-coated scaffolds were put in a desiccator to remove the solvent by vacuum at 10^−2^ mmHg for 4 h. Then, HA/PLA composite coating formed on the surface of 3D-printed PLA lines.

### 2.3. Thermal Gravimetric Analysis (TGA) of Scaffolds

To determine the amount of HA coated on PLA scaffold via dip coating, thermal gravimetric analysis was conducted with dried air purge on P-scaffolds before and after HA coating using a thermal gravimetric analyzer (TGA Q50, TA Instruments, New Castle, DE, USA). Before TGA test started, the sample was heated to 100 °C for 10 min to remove moisture in the furnace of the thermal gravimetric analyzer. Then, the TGA test was performed by heating the sample in dried air from 50 °C to 610 °C with a heating rate of 10 °C/min. In this study, the decomposition temperature (Td) of a sample was determined as the temperature at which 5% weight loss happened. The amount of HA was calculated according to the residual weight of samples that remained constant during the TGA test. 

### 2.4. Micro Structure Observation

A Leica DVM6 and a Leica DM2500 microscopes were used to observe the cross-section of the HA-coated P-scaffold. An HA-coated P-scaffold was polished to obtain the cross-section using 1000-grit sandpapers.

### 2.5. Animals

Two adult, male beagle dogs weighing 9 to 11 kg, about six months old, were used for the research. The animals had intact dentition with healthy periodontia. The animal research protocol was approved by the National Taiwan University College of Medicine and College of Public Health Institutional Animal Care and Use Committee (IACUC approval no. 20150320). 

### 2.6. Extraction Tooth and Sequential Fluorochrome Labeling

Tooth extraction was performed according to the reported method [17]. In brief, the animals were anesthetized using subcutaneous injection with a mixture of atropine (0.04 mg/kg), zoletil (10 mg/kg), and rompun (0.5 mg/kg) to reduce vomiting and maintain a regular heart rate. After 5–10 min, the animals were under general anesthesia. Local anesthesia was then administered on the surgical field by using 1.8 mL of Xylestesin-A containing epinephrine (1:100,000; 3 M ESPE, Seefeld, Germany). The animals were anesthetized, and the bilateral premolars (P2 and P4) and third incisor (I3) were then extracted using elevators and forceps. The wound was sutured using 3–0 absorbable sutures, and the stitches were removed for better hygiene care after 7 days. The outline of the experiment is shown in Figure 2. A polychrome sequential fluorochrome labeling method was performed in the experimental dogs, the animals were euthanized at week 40 after surgery to observe the new mineralization of tooth regeneration. For polychrome sequential fluorochrome labeling, each dog was injected by subcutaneous injections of calcein, alizarin and tetracycline, respectively (20, 30 and 25 mg/kg of body weight; Santa Cruz Biotechnology) at each time points of 0, 6, 12, 24, 30, 36 weeks after the surgery sequentially as was shown in Figure 2. Before use, all fluorochromes were adjusted to pH values of 7.2 and sterilized by filtration.

### 2.7. Cultivation of Dental Pulp Stem Cells of Dogs

Pulp tissues were isolated from the extracted teeth as follows. First, clean up the tooth surface with sterile PBS (Phosphate Buffered Saline), physically knock the tooth open to expose the pulp cavity, and carefully remove the pulp tissue from the pulp cavity with tweezers and stored in Dulbecco’s modified Eagle medium (DMEM, Chemicon, Rolling Meadows, Hurley, NY, USA) supplemented with 10% fetal bovine serum (FBS, Gibco-BRL Life Technologies, Waltham, MA, USA), 1% antibiotic/antimycotic (Gibco-BRL Life Technologies, Waltham, MA, USA) on ice. Subsequently, the pulp tissues were washed with Phosphate buffered saline (PBS) containing 2% antibiotic/antimycotic. After removal of the wash solution, pulp tissues were mixed with 500 µL DMEM medium cut into small fragments about 1 mm^3^ in size. The fragments were then placed in a 12-well cell culture plate and cultured at 37 °C in 5% CO_2_ atmosphere in a humidified incubator and changed medium every 2 days. Dental pulp stem cells of dogs (dDPSCs) were released from the pulp tissues and were grown to confluence in approximately 10–12 days. At approximately 90% confluence, pulp tissue fragments were removed and subcultured in 60-mm cell culture dishes (Corning, NY, USA) with fresh culture medium for one week. The total number of cells obtained from three extracted experimental teeth for primary culture was increased to approximately 3 × 10^5^ cells after 30 d in culture, and 1 × 10^5^ cells were loaded in each scaffold (I3, P2 and P4) separately for the experimental sides.

### 2.8. Surgical Procedures 

After the extraction sockets had healed (Figure 3a) for 4 weeks, the animals were subjected to a surgically created defect and 3D-printed HA/PLA transplantation. For each surgery, the alveolar defects of post-extraction regions of the incisor (I3) were created about 3 mm in diameter and 8 mm in depth separately on both sides of the mandible of each dog. The defects of post-extraction regions of premolars (P2 and P4) were created about 6 mm in diameter and 8 mm in depth and 6 mm × 8 mm separately in both sides of the mandible of each dog. The 3D-printed HA/PLA scaffolds with 3 mm in diameter and 8 mm in length were used for transplantation into the incisor defects, and the 3D-printed HA/PLA scaffolds with 6 mm in diameter and 8 mm in length were used for transplantation into the premolar defects (Figure 3b). For each dog, the left side (L-side) of the mandible was transplanted with 3D-printed HA/PLA scaffolds loaded with the dog’s dental pulp stem cells in the third incisor (I3), and two premolar areas (P2 and P4) were used as an experimental group. The right side (R-side) of the mandible transplanted with 3D-printed HA/PLA scaffolds without cells loading in the third incisor (I3) and two premolar areas (P2 and P4), were used as the control group. Primary wound closure was achieved with mattress and interrupted sutures using 5-0 Dexon and 4-0 Nylon [18]. In order to reduce the discomfort of the animals after the surgical procedures, both painkillers (PREVICOX (Firocoxib), 5 mg/kg) and antibiotics (cephalosporin, 12.5 mg/kg) were used. Furthermore, animals were fed with soft pet food pellets and canned pet food for one week after surgery.

### 2.9. Labeling 

#### 2.9.1. Cone Beam CT and Micro CT Observation

For dynamic observation of the mineralization for tooth regeneration, a dental cone-beam computerized tomography scanner (CBCT, Soredex, Minray, NY, USA; 120 kVp, 23.87 mAs) was used for assessment. For taking images, the dogs were anesthetized for investigation. Cone Beam CT images were taken for each dog before surgery and every month after surgery (Figure 4a,b). Forty weeks after the surgery, the animals were euthanized by intravenous injection of concentrated sodium pentobarbital (110–150 mg/kg of body weight). The mandibles were block-resected about 5 mm distance away from the spacing (around implant site) area and fixed in 10% formalin for further processing. For the calibration of the implant site, the radio-opaque gutta percha was placed in parallel positioned and imaged with microcomputed tomography, microcomputer tomography (micro-CT system, SkysScan 1076, Bruker, Billerica, MA, USA; 60 kV, 167 µA), through which the implant site was cut accurately for evaluation (Figure 4c).

#### 2.9.2. Undecalcified Ground Sections 

Specimens were prepared according to the previous reports [18]. In brief, each tissue block specimen was sectioned into two halves with a diamond disc (Isomet, Buehler, IL, USA) in a mesiodistal direction through the center and along the axis of the 3D-printed HA/PLA implants. One half was sequentially dehydrated in ascending concentrations of alcohols from 70% to 100%, infiltrated and embedded in methylmethacrylate resin. Following complete polymerization overnight, the specimens were cut to a size of less than 1 mm using a saw microtome (Leica SP1600, Buffalo Grove, IL, USA). The thin sections were then mounted on roughened glass slides and subsequently ground (PetroThin, Buehler, IL, USA) and polished (Minimet 1000, Buehler, IL, USA) to a final thickness of approximately 40 μm. 

### 2.10. Micro-CT and Fluorescence Analysis 

Micro-CT measurement was performed using a micro-CT system for quantifying mineralized tissue ingrowth inside the defect zones. To determine the amount and quality of the newly formed mineralized tissue over time, a 3-mm circular region of interest (ROI) inside the defect zones was chosen and three-dimensionally reconstructed using micro-CT Analyzer software (CT Analyzer). Fluorochrome labeling patterns were examined with a confocal laser scanning microscope (Carl Zeiss LSM880, White Plains, NY, USA) aided with appropriate filters. The dynamics of the new mineralization of tooth regeneration could be distinguished by green, red and, yellow fluorescence in accordance with calcein, alizarin and tetracycline administered at 0, 6, 12, 24, 30 and 36 weeks, respectively. In three images per sample, the areas occupied by each fluorochrome within the defect area. The percentages of calcein, alizarin and tetracycline fluorochrome labeling per sample were calculated using ImageJ software (NIH).

### 2.11. Decalcified Histology

The other half of each specimen was then demineralized in 14% (*w/w*) EDTA. The specimens were then dehydrated, embedded in paraffin, and sectioned (4-μm thick) for haematoxylin and eosin (HE) staining. All images were taken with Zeiss AxioImager (M1) scanning microscope.

### 2.12. Statistical Analysis

Results are presented as the mean (±standard deviation, SD). Statistical significance was calculated using one-way analysis of variance (ANOVA) followed by post-hoc procedure (Bonferroni analysis), with *p* < 0.005 considered significant for all tests.

## 3. Results

### 3.1. 3D-Printed HA/PLA Scaffolds

The printed I-scaffold and P-scaffold before and after dip-coating are shown in Figure 1c. The dimensions of printed scaffolds were similar to the design. The top ring and the root, mainly composed of root-circle and spacer, were printed. From Figure 1d, HA/PLA composite permeated the space between the printed PLA lines, so the 3D-printed PLA scaffolds were permeable as designed. However, the inserted image of Figure 1d shows the height of the oval cross-section, which was ~180 μm. Although the 100-μm gap between the two root-circles was not as well-formed as designed, there were still pores and gaps that remained and made the scaffold permeable. After dip-coating, the surface was covered by HA/PLA composite, and the pores and gaps between PLA lines were also filled. The 0.5 wt.% PLA in dip-coating solution acted as a binder that formed a composite layer with HA and the HA/PLA composite layer well adhered on PLA lines’ surface.

According to the results of TGA tests in Figure 1e, the residual weight of P-scaffold (3D-printed PLA), and both the top ring and root of the HA-coated P-scaffold was 0.01, 0.71, and 2.28 wt.%, respectively. The Td was 320.15, 301.95, and 267.47 °C for the 3D-printed P-scaffold and both the top ring and root of the HA-coated P-scaffold, respectively.

### 3.2. Clinical Findings

During the experimental period, we noticed animal body weight and measured it every six weeks. As shown in Figure 5a, animal body weight was increased after the experimental period. The experimental procedures did not cause any change in the behavior or eating habits of the dogs. Healing following tooth extraction was uneventful. The peri-implant mucosa at 0 weeks was free from clinical signs of inflammation (Figure 5b–d). All animals recovered well from the surgical procedures without significant complications after forty weeks of implanting. No signs of infection were noted (Figure 5b–d). Digital images of the buccal-lingual section of tooth engineered separated from beagle dogs were shown in Figure 5d. Both of the implant zones in the experimental, 3D-printed HA/PLA plus dDPSCs, and control, 3D-printed HA/PLA only groups, were similarly without infections after 40 weeks. 

### 3.3. Micro-CT and Fluorescence Analysis

After the animals were euthanized, the tissue section blocks were analyzed by micro-CT. New mineralized tissue in defect zones was measured by micro-CT analysis, as shown in Table 1. The basic morphometric indices include the measurement of mineralization tissue volume and the total volume of interest. The ratio of these two measures is termed mineralization tissue volume fraction. The experimental group (3D-printed HA/PLA plus dDPSCs) exhibited more mineralization tissue formation than the control group (HA/PLA) at week 40 with higher mineralization tissue volume fraction, mineralization tissue number, structure thickness, and lower structure model index. On the other hand, the structure separation in the two groups was almost the same. The 3D reconstructed measurement (Figure 6a) yielded the same results for the experimental group at Week 40 as the above micro-CT evaluation. The new mineralized tissue was distributed in the defect zones due to the infiltration and growth within interconnected pores of the 3D-printed HA/PLA scaffold (Figure 6a). The overlay images of the three fluorochromes were displayed in Figure 6b. Areas labelled with calcein (green), alizarin(red), and tetracycline(yellow) indicated mineralization tissue was formatted on the 3D-printed HA/PLA scaffold. Experimental and control groups showed limited fluorochrome incorporation. At week 40, the higher magnification of the fluorochrome area at defect zones (white boxes) areas was the incorporation of fluorescence in the experimental group was higher than that in the control group, indicating tissue mineralization occurred in the experimental group. The morphometric results of the fluorochrome-incorporated area in the experimental group showed higher mineralization than that in the control group (Figure 6d).

### 3.4. Histological Observations

Hematoxylin and eosin staining were performed to observe the microstructure of regenerated tissues. After 40 weeks, the spaces within the 3D-printed HA/PLA scaffold showed more mineralization tissue in the experimental groups (Figure 7). Compared to the control group implanted with scaffolds without cells at the P2 and P4 areas, the experimental group with 3D-printed HA/PLA scaffolds plus dDPSCs exhibited more mineralization tissue and showed much mature mineralized tissue in the spaces of scaffolds. The 3D-printed HA/PLA scaffolds possessed good biocompatibility because there was no evidence of inflammatory cell infiltration. 

## 4. Discussion

### 4.1. 3D-Printed Scaffolds

In this study, we designed liquid-permeable scaffolds with alternate stacking architecture of a root-circle and spacer. However, the height (~180 μm) and of extruded PLA lines were higher than that of our setting (100 μm in height), and thus, the gaps between two root-circle were less than 20 μm. Generally, the cross-section of extruded polymeric material lined is oval due to the cohesive force of melt polymeric materials before they cooled down to solidification. In addition, gravity makes the oval cross-section with a width higher than its height. The material flow of PLA has been controlled by the g-code created by the slicing program (Ultimaker Cura 2.7). Therefore, the designed cross-section area 2.50 × 104 μm^2^ (layer height of 100 μm × line width of 250 μm) could convert to the theoretical diameter of the round cross-section of extruded PLA line, which was 178.41 μm. While the PLA was just extruded from the hot nozzle to form the line, it had a round cross-section before it made contact with the previous underneath layer. Then, the extruded PLA was squeezed between the nozzle and the previous layer underneath due to the layer height being set as 100 μm. However, the height of the oval cross-section (~180 μm) was higher than the theoretical value (178.41 μm) of the PLA line with round cross-section, so the actual material flow during printing was larger than our setting. Thus, the gap and the space between the two root circles were compressed during 3D printing.

### 4.2. HA Coating on Scaffolds

According to the results of TGA tests, the amount of HA coated on the root and top ring of P-scaffold was 2.28 and 0.71 wt.%, respectively. The top ring was printed as a dense objective, so the HA/PLA composite was only coated on the surface. The designed root was composed of the alternate stacking architecture of spacer and root-circle, so it had larger surface area and more space than the top ring to coat HA/PLA composite. Although the alternate stacking architecture was not well-established during the 3D printing process, the scaffolds were liquid-permeable, as shown in Figure 1d, which demonstrates that the HA/PLA composite permeated into the space between two layers of PLA lines during dip-coating.

The 3D-printed PLA (P-scaffold without HA coating) showed good thermal stability in air and the Td of scaffold samples decreased while HA amount increased, as shown in Figure 1e. The weight loss from 100 to 200 °C of HA coated root and top ring was 1.67 and 1.07 wt.%. Before TGA test, the dip-coated scaffold samples were processed to remove solvent and moisture. The decrease of Td from the 3D-printed scaffold to HA coated root was 52.68 °C. Therefore, the weight loss should come from the solvent absorbed by the 3D-printed PLA lines. The Td decreased due to the absorbed solvent decreased the compactness of PLA lines and started to burn in air at low temperatures. The sample containing higher HA amount had higher surface area to contact and soaked the solvent, which resulted in decreased thermal stability. The solvent absorbed by biomaterials during processing might damage the loaded cell, the surrounding tissue, and the organ of the patients.

The dip-coated HA/PLA composite seemed to be biodegradable and was substituted by mineralized tissue, as shown in Figure 7. Originally, the liquid-permeable structures of 3D-printed PLA scaffolds were filled by HA/PLA composite after the dip-coating process. However, the degradation of the 3D-printed PLA line of scaffolds seemed to be very slow. The low in vitro degradation rate of 3D-printed neat PLA scaffolds in PBS at 37 °C has been reported, which was similar to our finding [19]. The glass transition temperature (Tg) of PLA is about 60 °C and the crystalline of PLA formed while the extruded hot melt PLA lines cooled down to solidification on the heated build plate of 60 °C. Both the high Tg and the crystalline of PLA hinder the degradation of PLA at a temperature below the Tg of PLA [20]. The dip-coated HA/PLA composite on the surface of 3D-printed PLA scaffolds was formed after being dried in a vacuum at room temperature, which might inhibit the crystallization of the PLA phase. The high content of HA in the HA/PLA composite increased the surface area of PLA phase. Thus, the dip-coated HA/PLA composite performed good biodegradability

This was the first study to challenge tooth regeneration in adult dogs with 3D-printed HA/PLA scaffolds loaded with dental pulp stem cells. In this study, an innovative and specially designed 3D-printed PLA scaffold with a similar root canal structure was used to measure tooth spacing and root size and then covered collagen and HA in an infiltration method to promote the differentiation of dental pulp stem cells. The size of the defect at the implantation site can be accurately drilled before the experiment was carried out so that the implant can be placed in the site very tightly and stably.

### 4.3. Evaluation

Before the experiment and after the operation, each dog was anesthetized first, and then CBCT was used to irradiate the implant site to observe the progress of tooth regeneration, which can be used for dynamic assessment.

The three-dimensional image of the craniofacial structure obtained through CBCT can avoid the problem of planar image magnification. After the digital development of dental imaging from the traditional film, the radiation exposure dose can be reduced by about 50–75%; however, the CBCT and traditional CT imaging principles are different. The latter uses a point source to obtain a cone-shaped beam after penetrating the object. For imaging on the film, CBCT uses slot technology, which was the X-ray from the source was passed through a linear slit to make the beam into a fan and then projected on the sensor for imaging. CBCT three-dimensional images can provide much diagnostic information, not only obtaining three-dimensional images but can also be further compressed into two-dimensional X-ray images that are normally used. The tube and the detector rotate to obtain projected images from various angles. Finally, the 3D image was reconstructed by the Feldkamp algorithm.

In this study, polychrome labeling with sequential fluorochrome was performed by subcutaneous injections of calcein, alizarin, and tetracycline, respectively, at each time point of 0, 6, 12, 24, 30, 36 weeks after the surgery sequentially. The technique of fluorochrome sequential labeling using calcium-binding fluorescent dyes was widely used for addressing different questions concerning the process of mineralization. It was simple and efficient for the investigation of the dynamics of mineralization in combination with plain histology [21]. Fluorochrome labels, when bound to calcium ions, can be incorporated at sites of mineralization in the form of hydroxyapatite crystals Ca_5_(PO_4_)_3_(OH) [22,23]. This means that the label indicates all sites of mineralization in the body, including a site of bone formation or dentin deposition, and also hypertrophic cartilage in the growth plate. In the first 24–36 h after administration, the label was stabilized. The unincorporated label was rapidly excreted by the kidneys, which resulted in a peak concentration in urine, ranging from 225 min in monkeys [24] to 30 min in mice [23]. As a result, the fluorescent label demarcates the mineralization front at the time of administration and can be detected in histological sections without any further staining or decalcification. By administrating different labels at specific time intervals, mineralization can be followed in time. In general, emission spectra of tetracyclines are in the yellow range (dimethyltin dichloride [DMTC] being orange-yellow), calcein green fluorochromes exhibiting a high calcium affinity label a broader fluorescent band. Calcein has a high calcium affinity which translates into a relatively broad fluorescent band [21,22,23,24]. Fluorochrome labeling allows the determination of the onset time and location of mineralization and the direction and speed of mineralization [25,26,27]. In this study, the method of intravenous injection of concentrated sodium pentobarbital for the sacrifice of dogs can effectively maintain the characteristics of the tissues, and it was not easy to cause deformation after sacrifice.

In this study, the implant site images were calibrated by using the radio-opaque gutta percha for parallel positioning, which can be used for accurately positioning and cutting the implant sites for evaluation.

Different techniques have been used to evaluate the mineralization tissue repair, as histomorphometry and morphometric analysis using micro-CT reconstructions. The basic morphometric indices include the measurement of mineralization tissue volume and the total volume of interest. The ratio of these two measures is termed mineralization tissue volume fraction [28].

One of the highlighted findings of the present study was the mineralization volume fraction and mineralization tissue numbers were higher on the HA/PLA+ dDPSCs groups than on the control (HA/PLA) groups. Mineralization tissue number, structure thickness, and structure separation were based on 3D calculations, namely, a sphere-fitting method, where for thickness measurement, the spheres are fitted to the object, and for separation, the spheres are fitted to the background [28]. The structure model index refers to the presence of plaques and rods in a 3D structure such as the mineralization tissue. Studies had used this index and revealed a negative correlation of structure model index with mineralization tissue volume fraction [29], compared with our results.

Micro-CT is a non-destructive 3D imaging technology that can clearly depict the internal microstructure of the sample without destroying the sample. There are three major hardware components, including the X-ray vacuum tube, the X-ray detection and recording device, and the device of the rotating mechanism.

The target object was first placed on the stage to be still, and the X-ray vacuum tube and the detector are driven to rotate by the rotating mechanism to obtain projection images from various angles. Finally, the 3D image was reconstructed by the Feldkamp algorithm. The biggest difference between it and ordinary clinical CT was its extremely high resolution. It uses a micro-focus X-ray tube that was different from ordinary clinical CT, which can reach the micrometer (μm) level. Micro-CT usually uses Cone Beam. Using cone beams not only can obtain truly isotropic volumetric images, improve spatial resolution, and increase ray utilization, but also the speed of acquiring the same 3D image was much faster than fan beam CT. Micro-CT can provide geometry information, including sample size, volume, and spatial coordinates of each point and structure information. Micro-CT can also provide material information such as attenuation value, density, and porosity of the sample, or mechanical parameters such as the elastic modulus, Poisson’s ratio, etc., analyze the stress and strain of the sample and perform non-destructive mechanical tests.

### 4.4. Clinical Findings

We found that the experimental group of implantations with dental pulp stem cells loaded on the scaffolds showed more mineralization than the control group of implantations without dental pulp stem cells. 

The dog’s dental pulp stem cells isolation and cultivation were performed according to our team’s previous experiences in the research of human dental pulp stem cells. After the dental pulp tissues were isolated, single-cell colonies were collected and subcultured.

For reducing the sacrifice of animals, only two dogs were used for the experiment. For each dog, three teeth (I3, P2 and P4) from both sides of the lower jaw were extracted for study; one side was used for the experimental group and the other side for the control group. In the experimental group, three defects in the left-lower jaw of each dog were transplanted with dental pulp stem cells loaded on 3D-printed HA/PLA scaffolds and have a total number of six. In the control group, three defects in the right lower jaw of each dog were transplanted with a 3D-printed HA/PLA scaffold without cells and had a total number of six. We compared the results of transplantation of 3D-printed HA/PLA scaffolds with and without dental pulp stem cells. Using fluorescent dyes injected in different periods can determine the amount of tissue mineralization accumulation so as to identify the effect of tissue mineralization regeneration. Of course, mineralized tissue does not represent teeth but only shows the differentiation potential of dental pulp stem cells. In the previous study, we used tooth germ cells to successfully regenerate tooth roots with pulp, dentin, cementum, and periodontal tissues, and the scaffold was completely absorbed [2], which may be due to the greater potential of tooth germ stem cells. Since it is not easy to obtain tooth germ cells in adults, this study was designed to investigate whether adult tooth missing can be regenerated with dental pulp stem cells that are easier to obtain from adults in real life. In this study, we found that the degradation of PLA was slow, so there was still some material remaining in the implantation site. However, it was clear that the addition of dental pulp stem cells can promote material degradation and the formation of mineralized tissue.

Previous research on the use of dental pulp stem cells for tooth regeneration has also shown that dental pulp stem cells have the potential to form tooth-like tissues by Gronthos’s group [4] and others [5,6,7,8]. When DPSCs were transplanted into immunocompromised mice, they generated a dentin-like structure lined with human odontoblast-like cells that surrounded a pulp-like interstitial tissue [4]. Other researchers also found the potential of DPSCs for dentinogenesis or dentin-pulp-like tissue regeneration but also only used mice as an animal model [5,8]. Our study is the first study using dogs as animal models for dental pulp stem cells tooth regeneration with a designed, 3D-printed root like an HA/PLA scaffold.

Advances in regenerative medicine with stem cells have led to clinical trials. Dental/oral tissues are emerging as promising cellular sources of human mesenchymal stem cells. Recently, dental tissue-derived cells have been used clinically due to their great potential, easy accessibility, and ability to be obtained via methods with low invasiveness [30]. Dental pulp stem cells (DPSCs) are the first discovered dental stem cells. DPSCs are ectoderm-derived stem cells isolated from the human permanent tooth pulp. These cells originate from migrating neural crest cells and have similar characteristics as mesenchymal stem cells (MSCs) [9]. They have already been shown to be capable of differentiation into several lineages, such as odontoblasts, osteocytes/osteoblasts, adipocytes, chondrocytes, and neural cells. DPSCs are peri-vascular located and express a number of mesenchymal stem cell (MSC) markers such as CD105, CD146, CD44, and Stro-1 [31]. The human tooth consists of enamel, pulp tissue, dentin, and cementum, and DPSCs have the capability to differentiate into all these four tissues, which make bio-tooth a potential reality.

Many studies have shown that cytokines/growth factors such as BMPs, FGFs, SHHs, WNTs and TNFs, play an important role during tooth development. Moreover, the expression of these cytokines is characterized by spatiotemporal specificity. The aberrant expression may lead to tooth development abnormalities. The spatiotemporal control of the developmental cues might be the future for tooth regeneration applications [1]. The 3D-printing technology is a rapid prototyping and additive manufacturing technology, which manufactures complex architecture via a layer-by-layer building process and with high precision [32]. The flexibility and controllability of 3D bioprinting enable complex and customized release profiles of multiple cytokines to achieve spatiotemporal gradients that regulate cellular functions in tissue or organ regeneration [33,34]. Moreover, many studies have promoted the application of 3D printing technology in cytokine sustained-release by improving processing, advancing technology, or allowing combinations with other forms of carriers [35,36,37]. Tooth regeneration with dental pulp stem cells and using more rapid degradable biomaterials for 3D printing scaffolds with cytokines/growth factors sustained-release effects should also be considered for further studies.

## 5. Conclusions

In this study, we challenged the use of dental pulp stem cells loaded in a 3D-printed HA/PLA scaffold for tooth regeneration research. It was found that after transplantation, the degree of mineralization in the experimental group with dental pulp stem cells seeded in the scaffolds was significantly higher than that of the control group transplanted with cell-free scaffolds. Dental pulp stem cells are important for the mineralization of tooth regeneration. It was also demonstrated that a 3D-printed HA/PLA scaffold designed as a root-like structure promotes mineralization of dental pulp stem cells; the use of polychrome labeling was helpful in exploring the process of mineralization. Dental CBCT also was important for dynamic observation of the mineralization process. However, the degradation rate of hydroxyapatite/polylactic acid was too slow and was not fully absorbed even after nine months. Tooth regeneration with dental pulp stem cells and using more rapid degradable biomaterials for 3D printing scaffolds with cytokine/growth factors sustained-release effects should also be considered for further studies. 

## Figures and Tables

**Figure 1 cells-10-03277-f001:**
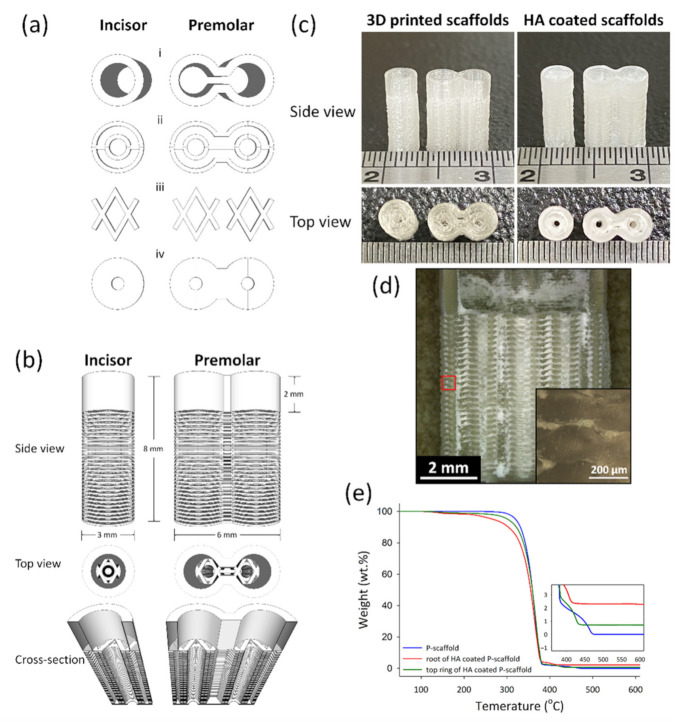
The architecture design of scaffolds and the 3D-printed scaffolds before and after HA coating. (**a**) The basic units to compose scaffolds: the top ring (i), root-circle (ii), spacer (iii) and bottom (iv). (**b**) The side view, the top view, and the cross-section of the designed scaffolds. (**c**) was the side view and top view of I-scaffold and P-scaffold before and after HA coating. (**d**) The cross-section of the HA-coated P-scaffold; the inserted image was the enlargement of the red square. (**e**) The TGA curves of 3D-printed P-scaffold and both the root and the top ring of the HA-coated P-scaffold.

**Figure 2 cells-10-03277-f002:**
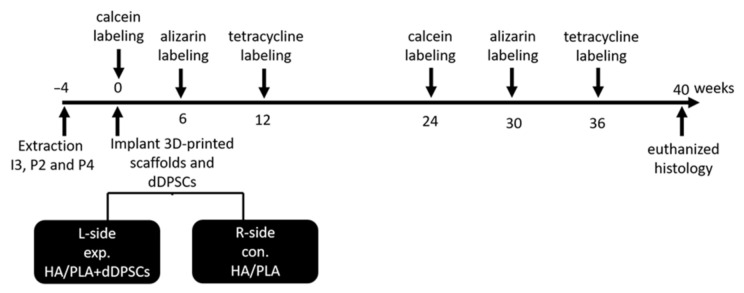
Outline of experiment. Animals *n* = 2. I3: the third incisor; P2: the second premolar; P4: the fourth premolar. exp: experimental; con: control. L-side: left side; R-side: right side.

**Figure 3 cells-10-03277-f003:**
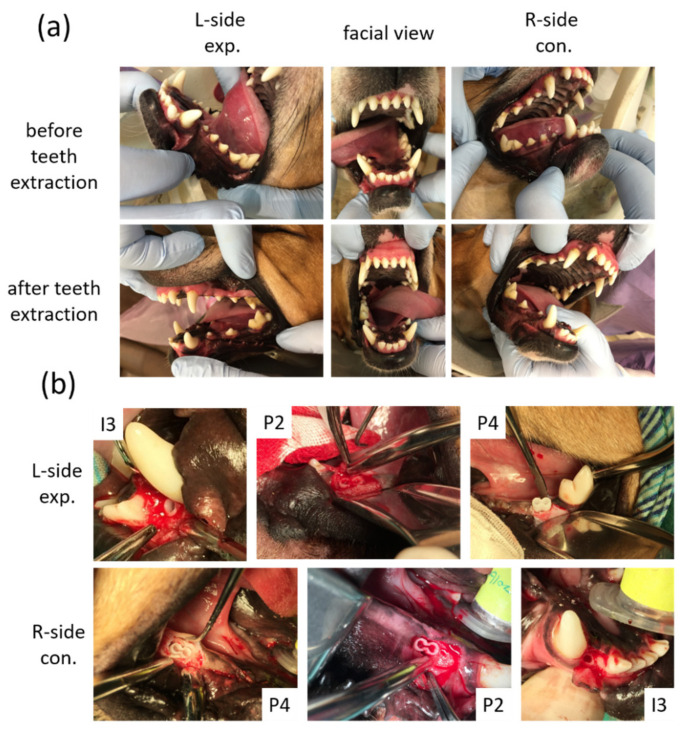
Macroscope observation for tooth regeneration. (**a**) The incisor and the premolar teeth in the right and left mandibular jaw quadrants before and after teeth extraction. (**b**) After implantation of 3D-printed HA/PLA scaffolds. L-side: left side; R-side: right side; I3: the third incisor; P2: the second premolar; P4: the fourth premolar.

**Figure 4 cells-10-03277-f004:**
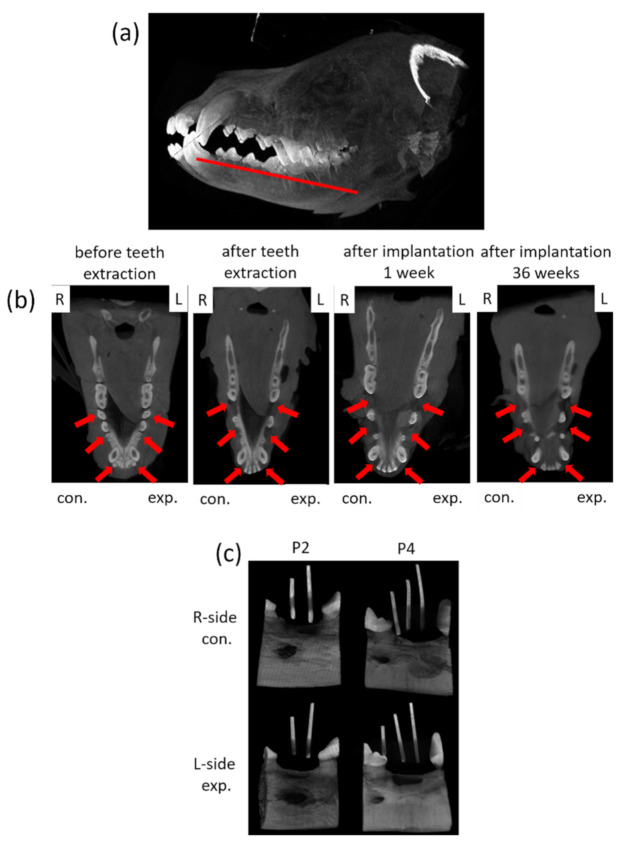
Photographs of the CBCT images and micro-CT observation of the tissue blocks. (**a**) The schematic of the observation section zone of the animal’s jawbone. (**b**) The mandible images of the animals were collected from CBCT before teeth extraction, after teeth extraction and after implantation for 36 weeks. L: left side, R: right side; con.: control, exp.: experimental; red arrow indicates the surgery area. (**c**) The micro-CT images of tissues blocks with radio-opaque gutta percha.

**Figure 5 cells-10-03277-f005:**
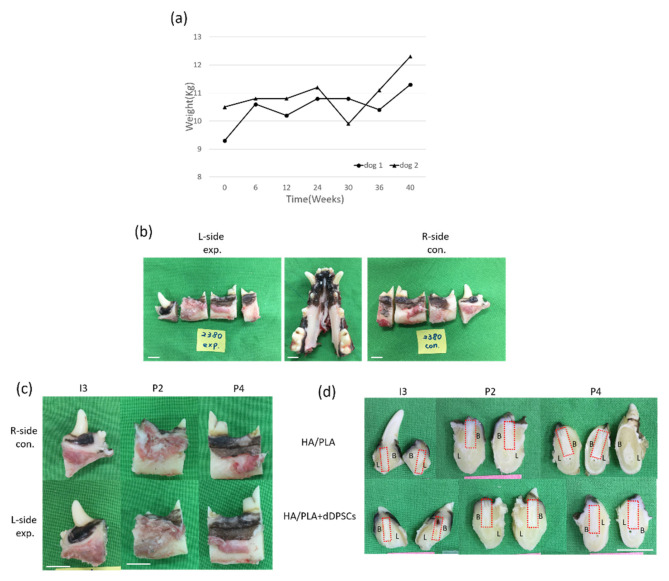
(**a**) The body weight of the experimental animals after the experiment. (**b**,**c**) Photographs from the tissues in dogs. Horizontal ramus of the dog mandible in lingual view. (**d**) Gross observation of buccal-lingual section of tooth engineered. L-side: left side; R-side: right side; I3: the third incisor; P2: the second premolar; P4: the fourth premolar. B: Buccal side; L: lingual side. Scale bar: 1 cm.

**Figure 6 cells-10-03277-f006:**
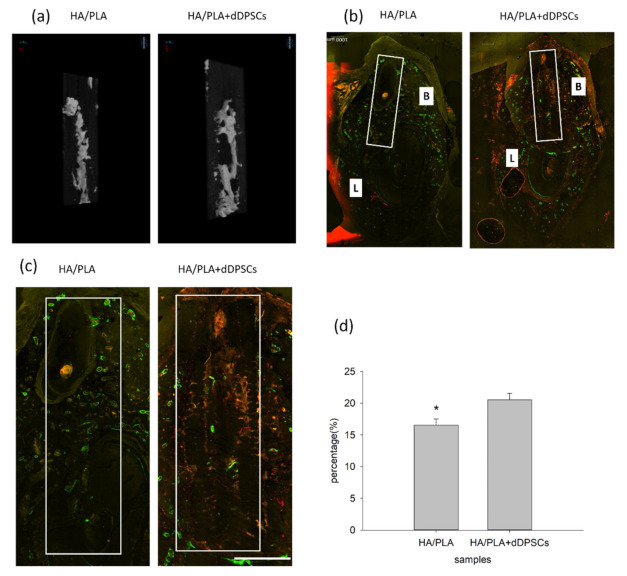
Micro-CT and merged confocal microscope images at 40 weeks. (**a**) 3D reconstruction micro-CT images of new formation mineralization zone. (**b**) Mesiodistal section view of merged confocal microscope images. The rectangle zone indicated the 3D-print HA/PLA scaffolds area. At week 40, the 3D-print HA/PLA plus dDPSCs group showed higher fluorescence intensity and more red fluorescence incorporation, which indicated tissue mineralization occurred at the defect zones. Scale bar: 1000 µm. (**c**) The higher magnification of the fluorochrome area at defect zones (white boxes) areas. Scale bar: 2 mm. (**d**) The percentage of fluorochrome area was calculated at defect zones (white boxes areas in the draw). * *p* < 0.05.

**Figure 7 cells-10-03277-f007:**
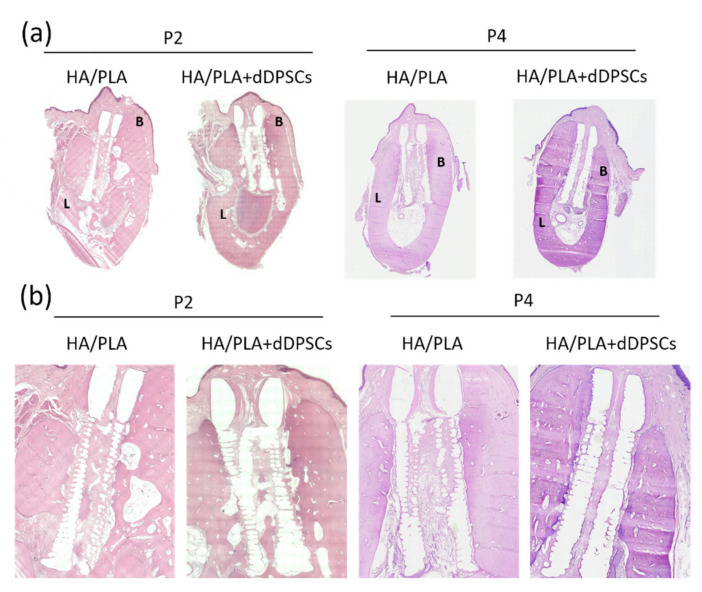
Photographs of the buccal-lingual histologic section of second and fourth premolar at 40 weeks. The (**a**) Mesiodistal section view. (**b**) Magnification. B: Buccal side; L: lingual side. Scale bar: 1 cm.

**Table 1 cells-10-03277-t001:** Micro-CT evaluation at 40 weeks.

**Mineralization tissue volume/total volume (%)**	10.79 ± 3.29	12.93 ± 2.43
**Mineralization tissue number (1/mm)**	0.69 ± 0.16	0.79 ± 0.12
**Structure thickness (mm)**	0.30 ± 0.05	0.32 ± 0.03
**Structure separation (mm)**	3.03 ± 0.58	3.03 ± 017
**Structure model index**	3.42 ± 0.80	3.27 ± 0.44

## Data Availability

The data that support the findings of this study are available from the corresponding author upon reasonable request.

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
