# Peer review of "Challenge Tooth Regeneration in Adult Dogs with Dental Pulp Stem Cells on 3D-Printed Hydroxyapatite/Polylactic Acid Scaffolds"

_cells, 2021, doi:10.3390/cells10123277_

Round 1
Reviewer 1 Report
Although the subject is relevant, the manuscript text lacks support from the previous literature and follows an unconventional format. Animal experiments are questionable from an ethical point of view due to their multiple surgery sites (beyond a second intervention). Moreover, accumulated evidence has shown that more than one surgical site may interfere in the global outcome.
This reviewer considered that this research over-extended the biomaterial limitations, thus compromising animal welfare and ethical barriers. Moreover, it presents minimal analysis and data extrapolation.
Some other concerns:
Inconsistency in the number of animals (2 or 3?)
MicroCT or CBTC?? Why both techniques?
Fluorescence analysis is unclear, so as its results.
There is no standard deviation or SEM in any data.
Which is the fluorochrome quantified in the graph (c) (Fig 6) - calcein, alizarin or tetracycline? Image quality is poor, higher magnification should be provided.
Replace sacrifice with euthanized
Lack of characterization studies for the scaffolds, resorbability in vitro tests
Data weaknesses and lack of originality
The conclusion is too obvious.
English is poor; acronyms without meaning.
Author Response
Reviewer 1
Although the subject is relevant, the manuscript text lacks support from the previous literature and follows an unconventional format. Animal experiments are questionable from an ethical point of view due to their multiple surgery sites (beyond a second intervention). Moreover, accumulated evidence has shown that more than one surgical site may interfere in the global outcome.
This reviewer considered that this research over-extended the biomaterial limitations, thus compromising animal welfare and ethical barriers. Moreover, it presents minimal analysis and data extrapolation.
Our response: Thanks for reviewer’s comments. To consider the replacement, reduction, and refinement (3R) of the animal experimental design. Animals used in this study were produced the compatible tooth extraction of the beagle dogs. Previous study (cited reference no. 10) had been reported, bilateral mandibular premolars were extracted from beagle dogs. In order to reduce the discomfort of the animals after the surgical procedures, both painkiller (PREVICOX (Firocoxib), 5 mg/KG) and antibiotic (cephalosporin, 12.5 mg/KG) were used. Furthermore, animals were fed with soft pet food pellets and canned pet food for one week after surgery. During the experimental period, we noticed animal body weight and measured that for every 6 weeks. As shown in the following figure, animal body weight was increased after the experimental period.
The body weight of the experimental animals after experiment.
Some other concerns:
- Inconsistency in the number of animals (2 or 3?)
Our response: Thanks for kindly reviewing the manuscript. These mistakes have been modified in the revised manuscript (page 5 lane 154). There were 2 animals in this study.
- MicroCT or CBCT?? Why both techniques?
Our response: Before the experiment and after the operation, each dog was anesthetized first, and then CBCT was used to irradiate the implant site to observe the progress of tooth regeneration every six months, which was used for dynamic assessment. After the animal euthanized, the tissue section blocks were investigated with by micro-CT. Micro-CT analysis provide more accurate results because of its much higher resolution irradiation (27.2 μm voxel size) than CBCT (greater than 200 μm voxel size) (ref. as shown below). However, high radiation exposure and the operation space of micro-CT scanner limits its clinical application for living animals. Hence, in our research the CBCT was used for dynamic observation of the mineralization during tooth regeneration period, micro-CT was used for calibration on the implant site and 3D reconstructed measurement at week 40 after animal euthanized.
Ref.: Taylor TT, Gans SI, Jones EM, Firestone AR, Johnston WM, Kim DG. Comparison of micro-CT and cone beam CT-based assessments for relative difference of grey level distribution in a human mandible. Dentomaxillofac Radiol. 2013; 42(3):25117764. doi: 10.1259/dmfr/25117764. Epub 2012 Sep 20. PMID: 22996393; PMCID: PMC3667532.
- Fluorescence analysis is unclear, so as its results.
Our response: Thanks for kindly reviewing the manuscript. The fluorescence analysis was rewritten in the materials and methods in page 9 lane 242. For fluorescence analysis, in 3 images per sample, the areas occupied by each fluorochrome within the defect area. The percentages of calcein, alizarin and tetracycline fluorochrome labeling per sample was calculated using ImageJ software (NIH) for fluorescence results,
- There is no standard deviation or SEM in any data.
Our response: Thanks for kindly reviewing the manuscript. These mistakes have been modified in the materials and methods section (page 9 lane 262) in the revised manuscript. For Statistical analysis, Results are presented as the mean (± standard deviation, SD). Statistical significance was calculated using one-way analysis of variance (ANOVA) followed by post hoc procedure (Bonferoni analysis), with p< 0.005 considered significant for all tests.
- Which is the fluorochrome quantified in the graph (c) (Fig 6) - calcein, alizarin or tetracycline? Image quality is poor, higher magnification should be provided.
Our response: Thanks for kindly reviewing the manuscript. The fluorochrome quantified in graph Fig. 6 (d) (originally, it was Fig. 6(c)) was calculated from 3 images per sample, the areas occupied by each fluorochrome within the defect area. The percentages of calcein, alizarin and tetracycline was calculated using ImageJ software (NIH)The higher magnification was shown in Figure 6 (c).
- Replace sacrifice with euthanized
Our response: Thanks for kindly reviewing the manuscript. These mistakes have been modified in the revised manuscript.
- Lack of characterization studies for the scaffolds, resorbability in vitro tests
Our response: Thanks for your kind advice. We have tested in vitro degradation rate of 3D printed PLA scaffold previously, and the degradation was very slow. The same phenomenon has also been observed, so we cited one paper (ref. 19) which has reported the results agree with our finding. The low degradation rate of “3D printed” PLA scaffold resulted from the high Tg of PLA and the crystalline formed during 3D printing process. We cited one more paper (ref. 20) which has presented the same point of view.
Ref.: [19] S.J. Lee, H.H. Jo, K.S. Lim, D. Lim, S. Lee, J.H. Lee, W.D. Kim, M.H. Jeong, J.Y. Lim, I.K. Kwon, Y. Jung, J.K. Park, S.A. Park, Heparin coating on 3D printed poly (l-lactic acid) biodegradable cardiovascular stent via mild surface modification approach for coronary artery implantation, Chemical Engineering Journal 378 (2019)[20] T. Serra, J.A. Planell, M. Navarro, High-resolution PLA-based composite scaffolds via 3-D printing technology, Acta Biomaterialia 9(3) (2013) 5521-5530.
- Data weaknesses and lack of originality
Our response: Thanks for kindly reviewing the manuscript. These mistakes have been modified in the revised manuscript.
- The conclusion is too obvious.
Our response: Thanks for kindly reviewing the manuscript. These mistakes have been modified in the revised manuscript.
- English is poor; acronyms without meaning.
Our response: Thanks for kindly reviewing the manuscript. These mistakes have been modified in the revised manuscript.

Reviewer 2 Report
The research design is interested and should make a clinical improvement from this study; however, significant improvement of the manuscript write-up should be addressed.
- References in introduction are mostly out of date, the author should expand it with more recent studies in dental pulp stem cells, tooth regeneration, and the scaffolds, etc.
- Did confocal microscopy performed on the dog? Or extracted the implant at each time point for confocal analysis?
- The author didn’t show standard deviations in the graph and table, neither performing any statistical analysis for the results, and thus one should not make any conclusion without significant differences.
- The manuscript is not well written, significant improvement should be made. The introduction is not well addressed the research purpose, and why did they decide to use 3D printed PLA/HA scaffold? The method is divided into many sections, but is not well-organized to understand the flow of experiments. Most of discussion could be merged into results, and the discussion part should be expanded, such as discussing this research with more clinical relevant studies or applications.
- PLA/HA material has been well studied as a biomaterial, its mechanical/ rheological properties and life time must have been published in many journals. The author should consider those factors and discuss in this study.
Author Response
Reviewer 2
The research design is interested and should make a clinical improvement from this study; however, significant improvement of the manuscript write-up should be addressed.
- References in introduction are mostly out of date, the author should expand it with more recent studies in dental pulp stem cells, tooth regeneration, and the scaffolds, etc.
Our response: Thanks for kindly reviewing the manuscript. These mistakes have been modified in the revised manuscript (page 1 lane 32).
- Did confocal microscopy performed on the dog? Or extracted the implant at each time point for confocal analysis?
Our response: Thanks for kindly reviewing the manuscript. In this study, we only used CBCT performed on the dog during the 40 weeks. Micro-CT and confocal microscopy were performed to observe the regenerated tissues section blocks after animals were euthanized.
- The author didn’t show standard deviations in the graph and table, neither performing any statistical analysis for the results, and thus one should not make any conclusion without significant differences.
Our response: These mistakes have been modified in the materials and methods section (page 9 lane 262)in revised manuscript. For Statistical analysis, Results are presented as the mean (± standard deviation, SD). Statistical significance was calculated using one-way analysis of variance (ANOVA) followed by post hoc procedure (Bonferoni analysis), with p< 0.005 considered significant for all tests.
- The manuscript is not well written, significant improvement should be made. The introduction is not well addressed the research purpose, and why did they decide to use 3D printed PLA/HA scaffold? The method is divided into many sections, but is not well-organized to understand the flow of experiments. Most of discussion could be merged into results, and the discussion part should be expanded, such as discussing this research with more clinical relevant studies or applications.
Our response: Thanks for you good suggestion. We added a section in INTRODUCTION to describe why we coated HA on 3D printed PLA scaffolds via dip-coating process (page 2 lane 59). Besides, we also discussed why the dip-coated layer of HA/PLA composite was biodegradable, but the 3D printed PLA scaffolds performed low biodegradability. We hope this study will provide a reference for the development of biomedical composite. For further clinical application for tooth regeneration.
- PLA/HA material has been well studied as a biomaterial, its mechanical/ rheological properties and life time must have been published in many journals. The author should consider those factors and discuss in this study.
Our response: Thanks for your kind advice. As compared with body temperature of human and animal, the glass transition temperature of PLA (~60 oC) is relatively high. Besides, PLA is a semi-crystalline polymer and the crystalline hinders the degradation at body temperature. We discussed why 3D printed HA/PLA scaffolds performed low biodegradability and cited two paper to support our results and findings (ref. 19 and 20).
Ref.: [19] S.J. Lee, H.H. Jo, K.S. Lim, D. Lim, S. Lee, J.H. Lee, W.D. Kim, M.H. Jeong, J.Y. Lim, I.K. Kwon, Y. Jung, J.K. Park, S.A. Park, Heparin coating on 3D printed poly (l-lactic acid) biodegradable cardiovascular stent via mild surface modification approach for coronary artery implantation, Chemical Engineering Journal 378 (2019)[20] T. Serra, J.A. Planell, M. Navarro, High-resolution PLA-based composite scaffolds via 3-D printing technology, Acta Biomaterialia 9(3) (2013) 5521-5530.
Reviewer 3 Report
The work addresses a current and relevant issue. PLA/HA scaffolds were produced by 3D printing, half of the scaffolds were enriched with dental pulp stem cells. The resulting implants/grafts were transplanted in 2 6-month-old beagle dogs. The authors examined mineralization by CBCT and polychrome sequential labelling, and microstructure was studied by H&E staining.
The work has 2 main shortcomings:
1. Detailed images are needed for histology and polychrome sequential labelling.
2. numerous typos prevent the reading flow.
Line 3 3D [space] [space] Printed, Hydroxyapatite
Line 5 Cross for equal contribution is missing
Line 16 hydroxyapatite, transplantation
Line 17 should be Polychrome sequential labelling
Line 20 hematoxylin
Line 29 Tooth regeneration was/is an important issue. This sentence is worded too simply and does not motivate the reader to read further. Why, for example, is the regeneration of teeth relevant?
Line 32 Dentin, labelling, hematoxylin should be written uniformly.
Line 33 tooth roots [missing space] [1] This must be corrected in the entire manuscript.
Line 53 3D printed
Line 68 equipped with a 250
Line 71 200°C
Line 82 Kimtech
Line 83 10-2
Line 88 was conducted
Line 90 and 92 100°C
Line 109 number [space] mg/kg
Line 118 Animals n=2
Line 121 [space] [space] First,
Line 127 (PBS) containing 2% antibiotic
Line 143-148 no space between number and mm unit.
Line 176 5 mm double space distance
Line 180 evaluation missing space (Figure 4)
Line 216 3D
Line 218 Figure 1c). The
Line 230 267.47°C
Line 243 c) [space] Gross
Line 245 1 cm The scale bar is missing in 5b)
Line 272 no scale bar
Line 285 Scale bar: 1 cm; the scale bar is missing in the figure itself.
Line 296 2.50 x 104 μm2
Line 316 and 319 200°C
Line 317 wt.%. Before TGA test,
Line 323 Missing word stability. , the solvent
Line 367 group [3] and others [4-7].
Line 401 calcein green fluorochromes
Line 409 gutta percha
Line 431 research. [Double space] It was found
Line 439 hydroxyapatite
Author Response
Reviewer 3
The work addresses a current and relevant issue. PLA/HA scaffolds were produced by 3D printing, half of the scaffolds were enriched with dental pulp stem cells. The resulting implants/grafts were transplanted in 2 6-month-old beagle dogs. The authors examined mineralization by CBCT and polychrome sequential labelling, and microstructure was studied by H&E staining.
The work has 2 main shortcomings:
- Detailed images are needed for histology and polychrome sequential labelling.
Our response: Thanks for kindly reviewing the manuscript. The higher magnification of polychrome labeling was shown in Figure 6 (c).
- Numerous typos prevent the reading flow.
Our response: Thanks for kindly reviewing the manuscript. These mistakes have been modified in the revised manuscript.
Line 3 3D [space] [space] Printed, Hydroxyapatite
Line 5 Cross for equal contribution is missing
Line 16 hydroxyapatite, transplantation
Line 17 should be Polychrome sequential labelling
Line 20 hematoxylin
Line 29 Tooth regeneration was/is an important issue. This sentence is worded too simply and does not motivate the reader to read further. Why, for example, is the regeneration of teeth relevant?
Line 32 Dentin, labelling, hematoxylin should be written uniformly.
Line 33 tooth roots [missing space] [1] This must be corrected in the entire manuscript.
Line 53 3D printed
Line 68 equipped with a 250
Line 71 200°C
Line 82 Kimtech
Line 83 10-2
Line 88 was conducted
Line 90 and 92 100°C
Line 109 number [space] mg/kg
Line 118 Animals n=2
Line 121 [space] [space] First,
Line 127 (PBS) containing 2% antibiotic
Line 143-148 no space between number and mm unit.
Line 176 5 mm double space distance
Line 180 evaluation missing space (Figure 4)
Line 216 3D
Line 218 Figure 1c). The
Line 230 267.47°C
Line 243 c) [space] Gross
Line 245 1 cm The scale bar is missing in 5b)
Line 272 no scale bar
Line 285 Scale bar: 1 cm; the scale bar is missing in the figure itself.
Line 296 2.50 x 104 μm2
Line 316 and 319 200°C
Line 317 wt.%. Before TGA test,
Line 323 Missing word stability. , the solvent
Line 367 group [3] and others [4-7].
Line 401 calcein green fluorochromes
Line 409 gutta percha
Line 431 research. [Double space] It was found
Line 439 hydroxyapatite
Round 2
Reviewer 2 Report
The authors have addressed the comments well in the manuscript; however, still lack of discussion about the results was shown in the manuscript. For example, what do those parameters mean in the table? Did the authors make significant breakthrough compared to other studies?
Author Response
Thanks for reviewer’s comments. We had rewritten the discussion of the parameters in the table in the revised manuscript (page 13 lane 437 and page 14 lane 498).
